# Inhibitor of FTO, Rhein, Restrains the Differentiation of Myoblasts and Delays Skeletal Muscle Regeneration

**DOI:** 10.3390/ani14162434

**Published:** 2024-08-22

**Authors:** Rongyang Li, Yan Cao, Wangjun Wu, Honglin Liu, Shiyong Xu

**Affiliations:** 1College of Animal Science and Food Engineering, Jinling Institute of Technology, Nanjing 210095, China; 2018205007@njau.edu.cn; 2Department of Animal Genetics, Breeding and Reproduction, College of Animal Science and Technology, Nanjing Agricultural University, Nanjing 210095, China; 2018205001@njau.edu.cn (Y.C.); wuwangjun2012@njau.edu.cn (W.W.); liuhonglin@njau.edu.cn (H.L.)

**Keywords:** m6A, Rhein, *FTO*, skeletal muscle, regeneration

## Abstract

**Simple Summary:**

N6-methyladenosine (m6A) impacts skeletal muscle development. Rhein, an anti-inflammatory agent, inhibits *FTO*, a key demethylase in m6A metabolism. Our study found that *FTO* and *ALKBH5* expression rises during skeletal muscle fiber formation, while m6A levels fall. After muscle injury, their expression and m6A levels initially surge and then decline. Rhein’s *FTO* inhibition reduces MyHC and *MyoG* expression, suppressing myoblast differentiation. In mice, Rhein decreased MyHC expression and muscle fiber area, delaying regeneration. Rhein’s ability to enhance m6A modification suggests a new medicinal use for this extract.

**Abstract:**

N6-methyladenosine (m6A) is a crucial RNA modification affecting skeletal muscle development. Rhein, an anti-inflammatory extract, inhibits *FTO*, a key demethylase in m6A metabolism. Our study showed that during muscle fiber formation, *FTO* and *ALKBH5* expression increased while m6A levels decreased. After muscle injury, *FTO* and *ALKBH5* expression initially rose but later fell, while m6A levels initially dropped and then recovered. Inhibition of *FTO* by Rhein reduced MyHC and *MyoG* expression, indicating myoblast differentiation suppression. In a mouse model, Rhein decreased MyHC expression and muscle fiber cross-sectional area, delaying muscle regeneration. Rhein’s ability to increase RNA m6A modification delays skeletal muscle remodeling post-injury, suggesting a new medicinal application for this plant extract.

## 1. Introduction

Rhein, a component of Rhubarb, is a natural compound isolated from fresh rhizomes of *Rheum coreanum* Nakai and has anti-inflammatory and anti-tumor activities [1]. It was reported that Rhein can significantly alleviate dextran sodium sulfate (DSS)-induced chronic colitis [2]. Uric acid was identified as a crucial modulator of colitis, whereas Rhein treatment led to decreased uric acid level. Rhein can block NADH dehydrogenase-2 activity to inhibit the growth of *C. acnes* [3], and has been reported as a novel histone deacetylase (HDAC) inhibitor with antifibrotic potency in human myocardial fibrosis [4]. Although Rhein has been extensively studied in tumor and inflammation, the role of Rhein in skeletal myogenesis and skeletal muscle regeneration is still unclear.

Regeneration of skeletal muscle is a complex process, and various factors are expressed in sequence and coordinated with each other to maintain muscle structure and function. The chronological progression of skeletal muscle regeneration encompasses four interlinked stages: extracellular calcium ion influx in large quantities after muscle fiber injury, proteolytic enhancement, rapid muscle fiber necrosis, and satellite cell migration to skeletal muscle in response to chemokine satellite cells’ (SCs’) migration to the site of injury [5]. Neutrophils and macrophages in skeletal muscles increase to remove necrotic tissue and prepare for muscle regeneration, and the immune cells, especially macrophages, can secrete some cytokines such as IGF-1 [6] and Meteorin-like (Metrnl) to promote the proliferation of satellite cells [7]. SCs are activated, proliferated, and eventually differentiated to form new muscle fibers with the nucleus in the middle. The repair of skeletal muscle is accompanied by the gradual maturation of new muscle fibers and remodeling of the extracellular matrix, regeneration of blood vessels, and restoration of skeletal muscle function. Muscle-specific adult stem cell (MuSC) differentiation and fusion is a highly ordered process, which is regulated by a serial of transcription factors, such as myoblast determination protein 1 (MyoD), myogenic factor 5 (Myf5), and myogenin (MyoG) [8]. Post-transcriptional modifications of these key regulators, mRNAs, may be critical for muscle fiber formation [9,10,11].

M6A modification, a type of post-transcriptional modification of RNA, has been widely studied in recent years. Methyltransferase-like 14 (METTL14), methyltransferase-like 3 (*METTL3*), and Wilms’ tumor 1-associating protein (*WTAP*) are some of the most extensively studied m6A writers, while YT521-B homology (YTH) domain proteins (YTHDF1, *YTHDF2*, YTHDF3, YTHDC1, YTHDC2) and eukaryotic translation initiation factor 3 subunit A (EIF3A) are among the most researched readers. AlkB homolog 5 (*ALKBH5*) and fat mass and obesity-associated protein (*FTO*) are erasers [12,13]. The process of mRNA demethylation is dynamic and reversible, and mediated by fat-mass-and-obesity-associated protein (*FTO*), which is the first discovered demethyltransferase [14]. Knockout or inactivation of *FTO* causes the increase of the m6A level by about 20%, and it was reported that *FTO* is required for myogenesis by positively regulating mTOR-PGC-1α-pathway-mediated mitochondria biogenesis [15]. Moreover, *FTO* directly targets Atg5 and Atg7 transcripts and mediates their expression in an m6A-dependent manner [16]. *FTO* deficiency decreases white fat mass and impairs ATG5- and ATG7-dependent autophagy in vivo. However, the role of *FTO* and the inhibitor Rhein in the process of skeletal muscle regeneration and muscle fiber formation is rarely reported.

In addition to Rhein’s anti-inflammatory and anti-tumor effects, it is also an inhibitor of *FTO* [17]. In the process of muscle fiber formation in vitro, the expression of *FTO* gradually increases, and the level of m6A decreases. However, there are few reports on the relationship between the formation of muscle fibers and the changes of m6A modification after skeletal muscle injury in vivo. Therefore, this study not only confirmed that the addition of Rhein to inhibit *FTO* can inhibit the production of muscle fibers in vitro, but also confirmed that Rhein can delay the regeneration of skeletal muscle after injury in vivo. Therefore, our research provides a new perspective for understanding the mechanism of Rhein in regulating skeletal muscle injury.

## 2. Materials and Methods

### 2.1. Animals

Male BALB/c mice aged 7–8 weeks were purchased from Nanjing Medical University, Nanjing. In each squirrel cage, 5 mice were raised in a room with a constant temperature (22 ± 2 °C), with a light/dark cycle of 12/12 h (lighting time was from 8:00 a.m. to 8:00 p.m.) and enough water and food provided. Rhein (Selleck, Shanghai, China) was administrated by oral gavage 50 mg/kg, after the 1.2% BaCl_2_ induction of tibialis anterior muscle injury, until the end of experiment.

### 2.2. Cell Culture and Cell Treatment

Mouse C2C12 cells were cultured in growth medium (GM), which consisted of 89% high glucose DMEM (Gibco, Grand Island, NY, USA), 10% (*v*/*v*) fetal bovine serum (Sigma-Aldrich, St. Louis, MO, USA), and 1% penicillin streptomycin (Gibco, Grand Island, NY, USA). The culture temperature was 37 °C, and the carbon dioxide concentration was 5%. The induced C2C12 differentiation medium (DM) was composed of 97% high-sugar DMEM, 2% horse serum (Gibco, Grand Island, NY, USA), and 1% penicillin–streptomycin. The C2C12 working concentration of Rhein (Abmole, Houston, TX, USA) was 10 μM and 20 μM [17].

### 2.3. qRT-PCR

Trizol reagent (Thermo Fisher Scientific, Waltham, MA, USA) was used to extract total RNA from mouse tissues and C2C12 cells. NanoDrop 2000 (Thermo Fisher Scientific) and denaturing gel electrophoresis were used to evaluate the RNA concentration and integrity. cDNA synthesis was performed using PrimeScript Real-Time Master Mix (Takara Biotech, Dalian, China). Quantitative real-time PCR (qRT-PCR) was conducted for mRNA detection, employing the AceQ qPCR SYBR Green Master Mix (Vazyme, Nanjing, China) on a Step One Plus real-time PCR platform. Table 1 outlines the primers utilized for the quantitative analysis. The relative expression levels were determined by applying the 2^−ΔΔCT^ method, with mouse *GAPDH* serving as the endogenous reference gene for normalization of the mRNA expression data.

### 2.4. Immunofluorescence

C2C12 cells were cultured in a 12-well culture plate in DM for 4 days with 10 μM and 20 μM Rhein. After that, the cells were washed 3 times with PBS, fixed in 4% paraformaldehyde for 30 min, and then washed 3 times with PBS. The cells were then incubated in 0.5% Triton X-100 for 15 min at −4 °C, and further washed 3 times. Subsequently, the cells underwent a 2 h blocking process at room temperature using a solution containing 1% bovine serum albumin. Following 3 washes with PBS, the cells were incubated overnight at 4 °C with MyHC antibody (1:50 dilution, sourced from the Developmental Research Hybridoma Bank, Iowa City, IA, USA) and *MyoG* antibody (1:100 dilution, ABclonal, Wuhan, China). After 3 additional washes, the cells were incubated in the dark with goat anti-mouse IgG (H + L) antibody conjugated to Alexa Fluor 488 (1:100 dilution, ZSGB-BIO, Beijing, China) and rhodamine (TRITC)-conjugated goat anti-rabbit IgG (H + L) antibody (also 1:100 dilution, ZSGB-BIO), each for 1 h at room temperature. Subsequent to 3 more washes, the nuclei were stained with DAPI in the dark. Finally, after a final series of 3 washes, images were captured using a confocal microscope. (LSM700META, Zeiss, Oberkochen, Germany).

### 2.5. Western Blotting

After the myoblasts were induced to differentiate in the differentiation medium for 4 days, the medium was discarded. We washed again with PBS and lysed with RIPA containing 1% PMSF for 10 min on ice. After high-speed centrifugation (12,000× *g*) at 4 °C for 10 min, the supernatant cells were collected for protein lysis. The protein concentration was quantified utilizing the BCA protein analysis kit (Beyotime, Shanghai, China). The protein samples were mixed with 5×SDS-PAGE sample loading buffer (Beyotime, Shanghai, China) and denatured at 99.9 °C for 10 min using a PCR instrument. The primary antibodies employed in this study included MyHC (1:200 dilution, Developmental Research Hybridoma Bank, Iowa City, IA, USA), *MyoG* (1:500 dilution, ABclonal, Wuhan, China), *FTO* (1:500 dilution, ABclonal, Wuhan, China), *ALKBH5* (1:2000 dilution, Proteintech, Wuhan, China), and *GAPDH* (1:2000 dilution, Cell Signaling Technology, Danvers, MA, USA). Protein expression was visualized using a horseradish peroxidase-conjugated anti-rabbit/goat IgG secondary antibody (1:5000 dilution, Cell Signaling Technology). Following SDS-PAGE separation, the proteins were transferred onto a PVDF membrane (Millipore Sigma, Billerica, MA, USA) for immunoblotting analysis with the aforementioned antibodies. The resulting images of the target proteins were captured using a chemiluminescence detection system from GE Healthcare, Piscataway, NJ, USA, and subsequently analyzed with ImageJ software (V1.8.0) for quantification (National Institutes of Health, Bethesda, MD, USA).

### 2.6. Hematoxylin and Eosin Staining and Tissue Immunofluorescence Staining

BaCl_2_ (1.2%) was injected into the tibialis anterior muscle to construct a skeletal muscle injury model [18]. The mouse tibial anterior muscles at 0, 1, 3, 5, 7, and 9 days of injury were collected for hematoxylin and eosin (H&E) stain; for the staining steps, we refer to the previous report [19]. The tibial anterior muscles derived from control and Rhein-treated groups on the 9th day of injury were collected and fixed with 4% paraformaldehyde for 24 h, and then dehydrated with 75%, 85%, 95%, and 100% ethanol gradients for 1 h before being embedded and sectioned.

After paraffin hydration, skeletal muscle tissue immunofluorescence staining was performed. The staining steps were the same as the cell immunofluorescence steps described above. The anti-N6 methyladenosine polyclonal antibody (Millipore, Burlington, MA, USA), anti-dystrophin polyclonal antibody (Proteintech, Wuhan, China), MyHC (1:200 dilution, Developmental Research Hybridoma Bank, Iowa City, IA, USA), and *MyoG* (1:500 dilution, ABclonal, Wuhan, China) were used as the primary antibodies, and the goat anti-rabbit IgG (H + L) antibody (dilution 1:100; ZSGB-Bio, Beijing, China) conjugated with Alexa Fluor 488 and rhodamine (TRITC)-conjugated goat incubate with anti-rabbit IgG (H + L) antibody (dilution 1:100; ZSGB-BIO, Beijing, China) were used as the secondary antibodies.

### 2.7. Detect m6A Levels

The EpiQuik m6A RNA Methylation Quantification Kit launched by Epigentek was used to detect the level of m6A [20]. For detailed steps, refer to the operation instructions.

### 2.8. Statistical Analysis

Statistical analysis was performed using Prism 8.0 software (GraphPad Software, San Diego, CA, USA). An unpaired two-tailed Student’s *t*-test was used to analyze the statistical significance between two groups. Data are expressed as mean ± SE unless otherwise noted, and the level of significance was set at *p* < 0.05.

## 3. Results

### 3.1. Dynamic Changes in m6A Erasers and m6A Levels during C2C12 Differentiation

The expression of *MyoG* was significantly increased during the C2C12 cells differentiation (Figure 1a), indicating that the induction of differentiation was successful in vitro. The mRNA and protein expression levels of two key genes of m6A demethylation, *FTO* and *ALKBH5*, were also significantly increased (Figure 1b–d). In contrast, the m6A level was decreased during the process of C2C12 cell differentiation (Figure 1e).

### 3.2. Dynamic Changes of m6A Erasers, Writers and Readers, and m6A Levels during Skeletal Muscle Injury

C2C12 differentiation models have confirmed that *FTO* in *ALKBH5* gradually increases during differentiation in vitro. However, skeletal muscle is also involved in the proliferation and differentiation of satellite cells in the process of injury. In order to further explore the relationship among the m6A erasers, writers and readers, and m6A levels in the skeletal muscle injury process, we constructed a skeletal muscle injury model by injecting 1.2% barium chloride (BaCl_2_) into the tibialis anterior muscle (Figure 2a). The results showed that the necrotic muscle fibers after injury were gradually eliminated by immune cells. After the third day, new muscle fibers were gradually formed, and the remodeling of muscle fibers was basically completed on D9 (day nine), indicating that the muscle injury model was successfully constructed in vivo. In addition, we detected changes in the mRNA levels of *FTO* and *ALKBH5* during skeletal muscle injury (Figure 2b,c). Interestingly, *FTO* increased significantly from the first day of injury (Figure 2b), and *ALKBH5* first increased and then decreased (Figure 2c). The protein level is consistent with the mRNA expression trend (Figure 2d). In addition, during the injury process, the level of m6A first decreased and then increased (Figure 2e).

Furthermore, we detected the changes in *METTL3*, *METTL14*, *WTAP* (Figure 3a), *YTHDF1*, *YTHDF2*, and *YTHDF3* (Figure 3b) by qRT-PCR, and the results showed that the expression for all these genes firstly increased and then decreased, suggesting that these genes may also be involved in regulating skeletal muscle regeneration.

### 3.3. Increase m6A Level by Inhibiting FTO Inhibit C2C12 Differentiation

The above results have confirmed that in the process of skeletal muscle injury, the expression levels of *FTO* and *ALKBH5* first increase and then decrease, and the methylation level of m6A first decreases and increases and then returns to normal levels. In order to further explore the effect of *FTO* on the formation of skeletal muscle fiber, we performed the *FTO* function inhibition experiment using its specific inhibitor Rhein in vitro. The results also showed that the formation of myotubes were significantly reduced in the 10 μM and 20 μM groups (Figure 4a). Immunofluorescence results showed that adding 10 μM Rhein significantly increased the level of m6A (Figure 4b). The qRT-PCR results showed that MyHC decreased significantly when treated with 10 μM and 20 μM Rhein (Figure 4c), while *MyoG* decreased significantly when treated with 20 μM Rhein (Figure 4d). The results of MyHC (Figure 4e,f) and *MyoG* (Figure 4g,h) immunofluorescence and Western blot (Figure 4i) also showed that 10 μM and 20 μM Rhein treatments inhibited the formation of myotubes.

### 3.4. Increase of m6A Level by Inhibiting FTO Delays the Regeneration of Skeletal Muscle after Injury

In order to investigate whether *FTO* regulates skeletal muscle regeneration, we further conducted the experiments in vivo. Specifically, 1.2% BaCl_2_ was injected into the tibialis anterior (TA) muscle of mice to construct the skeletal muscle injury model. At the same time, intragastric administration of Rhein was conducted from day one to eight. The experimental design is shown in Figure 5a. Our previous study had proved that the skeletal muscle damage in the control group was completed on day 9 [21]. Nine days after the injury, we found that the weight of the mouse skeletal muscle was significantly reduced (Figure 5b). The percentage of TA muscle in body weight decreased significantly (Figure 5c). M6A immunofluorescence staining also showed that the RNA methylation level of Rhein was higher than that of the control group (Figure 5d). In addition, the results of hematoxylin and eosin (H&E) staining showed that most of the nuclei in the intragastric Rhein group were still located in the center of the muscle fibers (Figure 5e), indicating that the muscle fibers were not fully mature. In the control group, the nuclei were primarily located on the side of the muscle fibers, meaning that most of them were mature muscle fibers. Dystrophin immunofluorescence results showed that the cross-sectional area of muscle fibers treated with Rhein was significantly reduced (Figure 5f,g). Western blot results showed that intragastric Rhein inhibits the formation of myosin heavy chain MyHC (Figure 5h,i). In summary, Rhein significantly delays the repair of skeletal muscle after injury.

## 4. Discussion

Some recent studies have confirmed that RNA methylation m6A is involved in the growth and development of skeletal muscle [9,21,22]. However, there are few reports on whether the RNA m6A methylation is involved in skeletal muscle diseases. This article confirms that in the process of muscle fiber formation, the desmethyltransferases *FTO* and *ALKBH5* gradually increase, leading to a gradual decrease in methylation. Furthermore, we found that in the process of skeletal muscle injury, the repair of skeletal muscle can be delayed by inhibiting the m6A demethylase *FTO*. Our research provides a new perspective for understanding the mechanism of Rhein in regulating skeletal muscle injury.

In the process of inducing myoblast differentiation in vitro, both *FTO* and *ALKBH5* gradually increased, causing a gradual decrease in overall m6A. In order to further prove that *FTO* is essential in the process of muscle fiber formation in vitro, the *FTO*-specific inhibitor Rhein was added to the medium. Adding Rhein can significantly increase m6A levels, which was consistent with previous study [17], and inhibit the differentiation of myoblasts. It is possible that some key genes have been modified by m6A during the formation of muscle fibers [23]. An article reported that the m6A modification level of goat skeletal muscle cells also gradually decreased during the differentiation process [24]. During the differentiation of goat myoblasts, GADD45B plays a crucial role. It reduces the amount of m6A modification, helps stabilize and enhance the expression of its own mRNA, and subsequently activates the p38 MAPK pathway. This activation, in turn, promotes the differentiation of the goat myoblasts. In addition, the level of autophagy increases during the formation of skeletal muscle fibers, and it has been reported that m6A is involved in the regulation of autophagy [25]. Therefore, we hypothesized that Rhein regulates muscle fiber formation by inhibiting related RNA m6A modification, and the specific mechanism still needs to be further explored. A study has reported that m6A demethylase *ALKBH5* drives denervation-induced muscle atrophy by targeting HDAC4 to activate FoxO3 signaling [26]. Whether *ALKBH5* and *FTO* collaboratively regulate the level of m6A, thereby controlling the formation of muscle fibers, is still worthy of further investigation.

In addition, the skeletal muscle injury model is actually a process of skeletal muscle regeneration and remodeling. When skeletal muscle is injured, satellite cells migrate to the injured site under the action of chemokines, and promote the proliferation of satellite cells and the formation of blood vessels under the action of some inflammatory factors [27]. After the satellite cells proliferate, they also further fuse to form new muscle fibers. In the process of skeletal muscle injury, *FTO* and *ALKBH5* first increased and then decreased. This is different from the biological functions of Rhein, such as anti-inflammatory, antioxidant, and anti-cancer [28]. The addition of *FTO* inhibitor Rhein significantly inhibits the regeneration of skeletal muscle and delays the remodeling of skeletal muscle.

## 5. Conclusions

In summary, inhibiting *FTO* activity can significantly inhibit the differentiation of myoblasts and inhibit the formation of muscle fibers. This study confirmed that Rhein differs from traditional anti-inflammatory function in that Rhein can delay the regeneration and remodeling of skeletal muscle after injury by increasing RNA m6A modification, providing a new theoretical basis for the medicinal use of Rhein.

## Figures and Tables

**Figure 1 animals-14-02434-f001:**
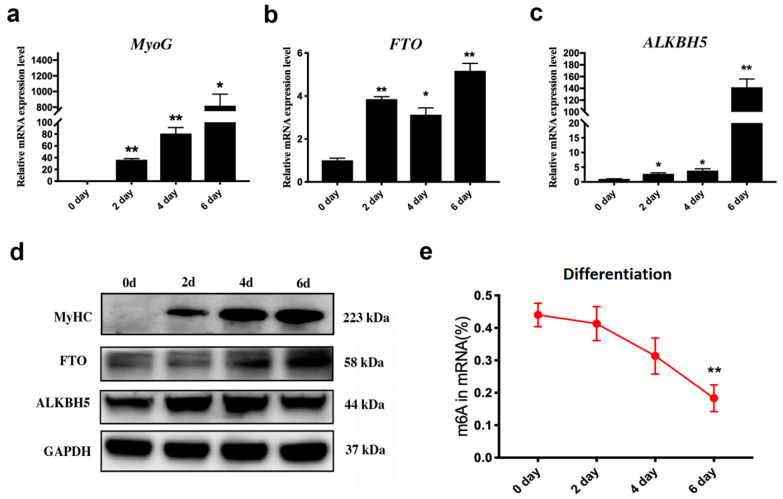
Changes in *FTO*, *ALKBH5*, and m6A levels during muscle fiber formation. The mRNA expression of *MyoG* (**a**), *FTO* (**b**), and *ALKBH5* (**c**) during C2C12 differentiation was analyzed by qRT-PCR. (**d**) Analysis of protein expression of *FTO* and *ALKBH5* during myoblast differentiation using Western blot. (**e**) The change in m6A levels during differentiation was detected by colorimetry (*n* = 3), * *p* < 0.05, ** *p* < 0.01.

**Figure 2 animals-14-02434-f002:**
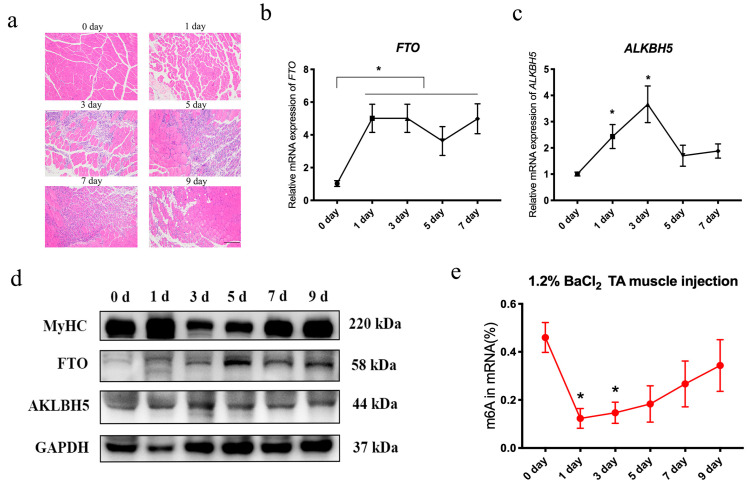
Changes in *FTO*, *ALKBH5*, and m6A levels during skeletal muscle injury. (**a**) Skeletal muscle injury model was confirmed after injection of 1.2% BaCl_2_ into tibialis anterior muscle by H&E staining; scale bars correspond to 100 μm (*n* = 3). The mRNA expression of *FTO* (**b**) and *ALKBH5* (**c**) during tibialis anterior muscle injury at 0, 1, 3, 5 and 7 days was analyzed by qRT-PCR (*n* = 3). (**d**) Western blot analysis of protein levels of MyHC, *FTO*, and *ALKBH5* during tibialis anterior muscle injury at 0, 1, 3, 5, 7, and 9 days. (**e**) The level of m6A during muscle injury was measured by colorimetry (*n* = 3). Data are expressed as mean values ± SEM, and an unpaired two-tailed Student’s *t*-test was used to analyze the statistical significance between two groups. * *p* < 0.05.

**Figure 3 animals-14-02434-f003:**
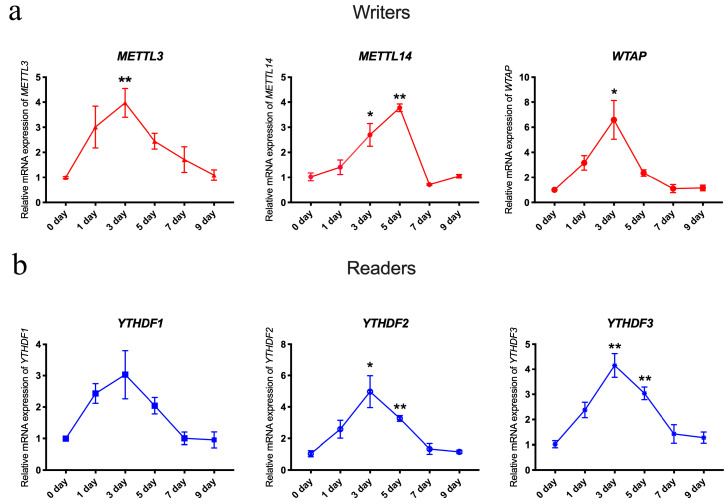
mRNA changes of m6A writers and readers during skeletal muscle injury. The changes in writers (*METTL3*, *METTL14*, *WTAP*) (**a**) and readers (*YTHDF1*, *YTHDF2*, *YTHDF3*) (**b**) were analyzed by qRT-PCR during days 0, 1, 3, 5, 7, and 9 (*n* = 3). Data are expressed as mean values ± SEM, and an unpaired two-tailed Student’s *t*-test was used to analyze the statistical significance between two groups. * *p* < 0.05, ** *p* < 0.01.

**Figure 4 animals-14-02434-f004:**
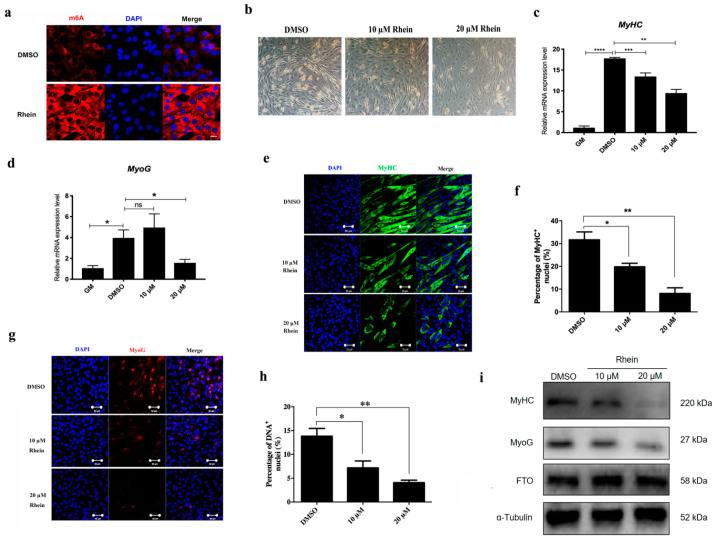
Addition of Rhein inhibits myoblast differentiation in vitro. (**a**) C2C12 was treated by Rhein of 10 μM and 20 μM, respectively, under bright field conditions and induced differentiation of myoblasts for 3 days; scale bars correspond to 100 μm. (**b**) Changes in m6A in myoblasts treated with 10 μM Rhein for 24 h were detected by immunofluorescence staining; scale bars correspond to 20 μm. The mRNA expression of MyHC (**c**) and *MyoG* (**d**) after treatment with 10 μM and 20 μM for 24 h was analyzed by qRT-PCR. The expression of MyHC (**e**) and *MyoG* (**g**) in C2C12 treated with 10 μM and 20 μM Rhein was detected by immunofluorescence staining; scale bars correspond to 50 μm. The immunofluorescence results of MyHC (**f**) and *MyoG* (**h**) were quantified. (**i**) Western blot analysis of MyHC, *MyoG*, and *FTO*. C2C12 myoblasts were treated with 10 μM and 20 μM Rhein for 3 days after induction and differentiation. Data are expressed as mean values ± SEM, and an unpaired two-tailed Student’s *t*-test was used to analyze the statistical significance between two groups. * *p* < 0.05, ** *p* < 0.01, *** *p* < 0.001, **** *p* < 0.0001.

**Figure 5 animals-14-02434-f005:**
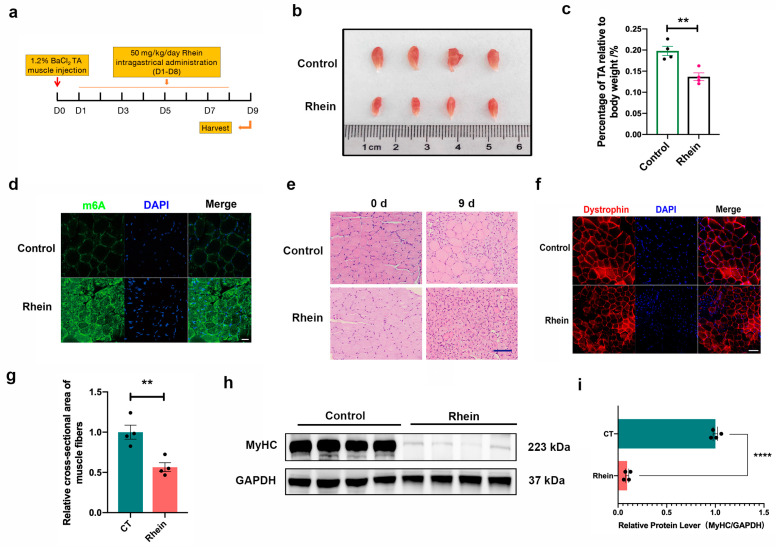
Inhibition of *FTO* activity hinders skeletal muscle repair after injury. (**a**) Experimental design of tibialis anterior muscle injection with 1.2% BaCl_2_ and Rhein intragastric administration. (**b**) Tibialis anterior on day 9 after injury. (**c**) Statistical analysis of percentage of body weight of tibialis anterior muscle (*n* = 4). (**d**) The level of m6A in the tibialis anterior muscle was analyzed by immunofluorescence staining on day 9 of injury. The scale is 20 μm. (**e**) H&E staining was used to analyze the repair process of the tibialis anterior muscle on day 9 of injury. The scale is 50 μm. (**f**) Dystrophin immunofluorescence staining was used to analyze the cross-sectional area of tibialis anterior muscle fibers on day 9 of injury. The scale is 50 μm. (**g**) The mean cross-sectional area of muscle fibers was quantified (*n* = 4). (**h**) Western blot analysis of MyHC expression on day 9 of tibialis anterior muscle injury. (**i**) Image J was used for quantitative analysis of Western blot results (*n* = 4). Data are expressed as mean values ± SEM, and an unpaired two-tailed Student’s *t*-test was used to analyze the statistical significance between two groups., ** *p* < 0.01, **** *p* < 0.0001.

**Table 1 animals-14-02434-t001:** Primer sequences for mRNAs.

Gene Name	Forward Primer	Reverse Primer
*MyoG*	GAGACATCCCCCTATTTCTACCA	GCTCAGTCCGCTCATAGCC
*FTO*	CCGAGTCGTCCGGACTTTAC	GGAAACCACGTCTGTGAGGT
*ALKBH5*	GACAACTACAAGGCGGGCA	CCAGAGGCATACAGGGAC
*METTL3*	GGGCACTTGGATTTAAGGAACC	CTTAGGGCCGCTAGAGGTAGG
*METTL14*	GAGCTGAGAGTGCGGATAGC	GCAGATGTATCATAGGAAGCCC
*WTAP*	ATGGCACGGGATGAGTTAATTC	TTCCCTTAAACCAGTCACATCG
*YTHDF1*	ACAGTTACCCCTCGATGAGTG	GGTAGTGAGATACGGGATGGGA
*YTHDF2*	GAGCAGAGACCAAAAGGTCAAG	CTGTGGGCTCAAGTAAGGTTC
*YTHDF3*	GATCAGCCTATGCCATATCTGAC	CCCCTGGTTGACTAAAAACACC
*GAPDH*	ATCACTGCCACCCAGAAGACT	CATGCCAGTGAGCTTCCCGTT

## Data Availability

The datasets used and analyzed during the current study are available from the corresponding author on request.

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
