# Peer review of "Inhibitor of FTO, Rhein, Restrains the Differentiation of Myoblasts and Delays Skeletal Muscle Regeneration"

_animals, 2024, doi:10.3390/ani14162434_

Round 1

Reviewer 1 Report

Comments and Suggestions for Authors

As the Authors resubmitted virtually the same paper, I did only change the line numbers if necesary.

Comments on the Quality of English Language

The paper requires English editing and rewriting, in many places the sentences are not clear. 

Author Response

Dear Editor,

I hope this message finds you well. I am writing to inform you that we have diligently addressed all the comments and suggestions made by both the academic editor and the reviewers.

We have made corresponding modifications in response to each of the comments and suggestions provided by the academic editor and reviewers. The following content is divided into two parts: Part 1 mainly involves the general comments from the reviewers, to which we have offered point-to-point responses. Part 2 specifically addresses each question raised by the academic editor with detailed clarifications. In addition, we have changed the order of the author's institutions. We are truly grateful for your guidance and look forward to your further review.

Part 1

General comments

  1. The paper requires English editing and rewriting, in many places the sentences are not clear

(line 39 to 42, line 53, line 66-67, line 180 – 182, line 250 – 251, line 289, line 299 – 301, line 302 – 305). Authors use the same names with both a capital letter or a lowercase letter, for example rhein (line 35, both cases!). What Metml (line 45) means?

Response:Thank you very much. I have made the changes and highlighted them in red in the revised manuscript. "rhein" has been uniformly changed to "Rhein", and "Meteorin-like (Metrnl), a cytokine, as a critical regulator of muscle regeneration" has been annotated in the text.

  1. In the Introduction some more information about mA6 modification should be provided, especially about mA6 writers, erasers and redears (these factors appear in Results). Authors are writing that ‘our data provides potential therapeutic targets for the treatment of skeletal muscle injury and skeletal muscle atrophy, and a new theoretical basis for the medicinal application of rhein’, but this statement is not discussed in more detail in the paper. 

Response:Thank you very much. I have further improved the introduction of m6A writers, erasers, and readers in the preface. And I have also re-described the significance of this work highlighted them in red in the revised manuscript.

  1. The study appears to have appropriate methodology, however authors should provide the full names for analyzed genes (2.3 qRT-PCR, Table 1).

Response:Thank you for your suggestion. I have written the full name of the gene for the first time it appears in the text, and then abbreviated it. These genes are also very common.

  1. The descriptions of Western Blotting (subsection 2.5) and H&E and IF staining (subsection 2.6)must be rewritten, as the subsequent steps of lab procedures are mixed.

Response:Thank you very much. I have revised sections 2.5 and 2.6 and highlighted the changes in red.

  1. The information about the number of used animals in skeletal muscle injury model is missing.

Response: Thank you very much. I have indicated the number of animals used in the experiments related to skeletal muscle injury.

  1. The number of replications considering the experiments using the mouse cell line is also not provided.

Response:Thank you very much. We did include cell replicate experiments in the original WB bands files we submitted.

  1. In the Results section the Western blotting results should be analyzed quantitatively (Fig. 1,

Fig. 2, Fig. 4). It is done only in Fig. 5.

Response:Thank you for your suggestion. We have conducted repeat experiments for Western blotting in Figure 1, Figure 2, and Figure 4 to observe the trend of protein expression, while quantification in Figure 5 is to indicate the number of experimental animals. Therefore, we did not quantify the Western blotting results in Figure 1, Figure 2, and Figure 4.

  1. Discussion section, especially the last part, must be rewritten to be more relevant to the

results.

Response:Thank you for your suggestion. I have revised the discussion section and highlighted the changes in red.

Part 2

  1. The academic editor has checked the original western blot images and raised some concerns as follows:

----------------------------------------------------

"a. In all original images, the markers were not shown.

Response:Thank you very much for your suggestion. At that time, we did not use a visible marker, and due to the limitations of the instrument, the marker bands did not show up in the pictures. In future experiments, we will use a marker that can be imaged. Thank you again for your advice.

  1. In Fig.1d:the bands for MyHC at 2d in original image are different between repeat 1 and repeat 2, and in the used repeat 1, just partial band presented, maybe this result is not representative.

Response:Thank you very much for your suggestion. The system for inducing C2C12 to form myotubes is very mature in our lab. Both repeat 1 and repeat 2 in MyHC figure 2d can prove that the induction of C2C12 to form myotubes is successful in vitro (with gradually increasing MyHC expression), and they are both representative.

  1. In Fig. 4i:the bands for FTO and α-Tubulin in original images were

artificially elongated and thinned in Fig. 4i."

Response:Thank you very much for your suggestion. I have readjusted the width of the bands to an appropriate size, and in the latest revised manuscript, I have highlighted the changes in red font.

---------------------------------------------------

2.We would be grateful if you could provide us with the corresponding

information at your earliest convenience.

Response:Thank you very much for your suggestion. Here is the relevant information about the ethics committee and approval.

Ethic Committee Name: Laboratory Animal Welfare and Ethics Committee of Nanjing Agricultural University.

Approval Code:NJAU.No20190715N25

Approval Date: July 15th, 2019

  1. We noticed that you uploaded Figure 6 to the Graphical Abstract section. Please confirm whether this image is uploaded to the system as the original image of Figure 6, or if it is intended to be published as a graphic abstract.Note: The GA should not be the same as a Figure in the manuscript.

Response:Thank you very much for your check. We have decided to use Figure 6 as the Graphical Abstract, and it will no longer appear in the manuscript.

  1. Please kindly reduce the similarity (attachment) during revision.

Response:Thank you very much for your verification. I have revised the parts with high repetition rate and highlighted them in red.

  1. Please complete the backmatter (attachment) during revision.

Response:Thank you very much for your suggestion. I have improved the backmatter as required.

  1. Ensure all references are relevant to the content of the manuscript.

Response:Thank you very much for your suggestion. All the references are relevant to the content of the manuscript.

  1. Highlight any revisions to the manuscript, so editors and reviewers

can see any changes made.

Response:Thank you very much for your suggestion. I have highlighted the revised parts in red.

Reviewer 2 Report

Comments and Suggestions for Authors

1. Title is not easy to understand. It could be revised as "Rhein (Inhibitor of FTO) Restrains the Differentiation of Myoblasts and Delays Skeletal Muscle Regeneration".

2. "Simple summary" needs to be rewritten. It's supposed to be easily understood by general people. So many abbreviations in simple summary is inappropriate.

3. Line 181-182: reword. Do you mean "However, the proliferation and differentiation of satellite cells are involved in the injury process of skeletal muscle.“?  

why the authors didn't detect the proliferation and differentiation of satellite cells, as they stated.

4. check the grammar of the subtitles 3.3 and 3.4.

5. Discussion needs to be deepened based on the current data.

6. The abbreviations in Abstract needs to be defined.

Skeletal muscle regeneration/repair is a complex process in which multiple factors (such as mRNA modification m6A) are involved. Rhein (a natural component from rhizomes of Rheum Coreanum Nakai) has been extensively studied for its effects in anti-tumor and inflammation.

However, its role in skeletal myogenesis and muscle regeneration is still unclear. This manuscript tackled this question. The authors demonstrated that rhein can inhibit fat mass and obesity- associated protein (FTO, a key demethylase in m6A metabolism), thereby negatively affects the regeneration/repair of skeletal muscle. Accordingly, Rhein can not be used for the treatment of skeletal muscle injury/atrophy. The reviewer asked the authors further explain the theoretical basis meaning of rhein medicinal application.

Comments on the Quality of English Language

Overall the quality of English is good. But improvements still need to be made by careful checking the whole manuscript.

Author Response

(The authors gave the same response as above.)
